# Ceramic-Polymer Composite Membranes for Water and Wastewater Treatment: Bridging the Big Gap between Ceramics and Polymers

**DOI:** 10.3390/molecules26113331

**Published:** 2021-06-01

**Authors:** Masashi Kotobuki, Qilin Gu, Lei Zhang, John Wang

**Affiliations:** Department of Materials Science and Engineering, National University of Singapore, 9 Engineering Drive 1, Singapore 117575, Singapore; msemk@nus.edu.sg (M.K.); msegq@nus.edu.sg (Q.G.); msezlei@nus.edu.sg (L.Z.)

**Keywords:** composite membrane, wastewater treatment, polymeric membrane, ceramic membrane, nanocomposite

## Abstract

Clean water supply is an essential element for the entire sustainable human society, and the economic and technology development. Membrane filtration for water and wastewater treatments is the premier choice due to its high energy efficiency and effectiveness, where the separation is performed by passing water molecules through purposely tuned pores of membranes selectively without phase change and additional chemicals. Ceramics and polymers are two main candidate materials for membranes, where the majority has been made of polymeric materials, due to the low cost, easy processing, and tunability in pore configurations. In contrast, ceramic membranes have much better performance, extra-long service life, mechanical robustness, and high thermal and chemical stabilities, and they have also been applied in gas, petrochemical, food-beverage, and pharmaceutical industries, where most of polymeric membranes cannot perform properly. However, one of the main drawbacks of ceramic membranes is the high manufacturing cost, which is about three to five times higher than that of common polymeric types. To fill the large gap between the competing ceramic and polymeric membranes, one apparent solution is to develop a ceramic-polymer composite type. Indeed, the properly engineered ceramic-polymer composite membranes are able to integrate the advantages of both ceramic and polymeric materials together, providing improvement in membrane performance for efficient separation, raised life span and additional functionalities. In this overview, we first thoroughly examine three types of ceramic-polymer composite membranes, (i) ceramics in polymer membranes (nanocomposite membranes), (ii) thin film nanocomposite (TFN) membranes, and (iii) ceramic-supported polymer membranes. In the past decade, great progress has been made in improving the compatibility between ceramics and polymers, while the synergy between them has been among the main pursuits, especially in the development of the high performing nanocomposite membranes for water and wastewater treatment at lowered manufacturing cost. By looking into strategies to improve the compatibility among ceramic and polymeric components, we will conclude with briefing on the perspectives and challenges for the future development of the composite membranes.

## 1. Introduction

In pace with the ever rapid urbanization, industrialization, and population growth, the water shortage, especially clean water, has become one of the most critical problems globally. There is also a concurrent environmental issue, if and when untreated wastewater is released into rivers, lakes and seas directly, leading to surface and ground water contamination and depletion of the clean water supply [1]. Moreover, the ever rapid growing population in many regions has increased the demand for freshwater supply more recently. The United Nations has predicted that half of the countries worldwide will suffer from water shortage in the coming decade [2]. Therefore, water and wastewater treatments have been considered to be an essential and critical factor for sustainable human society development.

Several treatment processes can be used to treat wastewater and supply clean water. For example, membrane separation, adsorption, chemical precipitation, electrochemical treatment, ion-exchange, chlorination, and ozonation, etc., are among the commonly used techniques to remove hazardous materials for clean water supply [3]. Among them, membrane separation has been the most widely used, due to its high energy efficiency and relatively low cost. The membrane separation is performed by passing desired molecules through pores in membranes selectively based on their sizes, and the interaction between the molecules and the pores without phase change and additional chemicals [4]. The membrane systems are generally simple and consist of limited ancillary equipment. Due to the low cost and high efficiency, membrane technology has been widely employed for the production of various types of water [5], dialysis of blood and urine [6], ion separation in the electrochemical processes [7], and filtration of particulates from liquid suspensions [8].

The separation through membranes occurs via three different mechanisms: (i) size-exclusion induced by the pores across the membrane, which allows passage of compounds smaller than the pore size, (ii) pore flow caused by the interaction between pore surface and passing molecules, which induces selective transportation of molecules with similar size to the pores, (iii) solution diffusion induced by the diffusion of molecules into the membrane, resulting in migration of the molecules across the membrane, which occurs exclusively in polymeric membranes. The fundamental properties that determine membrane performance are mainly the flux rate, selectivity, mechanical/chemical/thermal stabilities under operating conditions, fouling properties, and service durability. Since the membranes act as a barrier in the separation process, their surface properties, such as pore size, pore structure, surface roughness, and physicochemical properties largely influence the overall membrane performance. The roughness and hydrophilicity of the membranes influence the fouling behavior of the membrane significantly, while the mechanical/chemical/thermal stabilities determine the lifespan of the membranes [9].

Pore size, pore size distribution, morphology and surface properties of membranes primarily define membrane properties. For example, the pore size and its distribution mainly determine the selectivity, while properties of membrane materials influence permeability, fouling and selectivity. The overall thickness and pore shape in the membranes also affects the membrane flux. According to the pore size range, the membrane separation process for water purification and desalination can be classified into microfiltration (MF), ultrafiltration (UF), nanofiltration (NF) and reverse osmosis (RO) (Figure 1) [10]. In macroporous MF and mesoporous UF membranes, the principle phenomenon of solute rejection is the molecular sieving/size exclusion mechanism [11,12]. MF membranes typically reject those suspended particles, asbestos, and cellular materials, such as particles, bacteria, protozo, a and red blood cells. Contrarily, UF membranes have smaller pores and can reject smaller particles, microsolutes such as sugars and salts and macromolecules such as pyrogens, proteins, and viruses. NF membranes can reject low molecular weight-uncharged solutes by the size exclusion mechanism such as MF and UF membranes and charged molecules by a combination of the size exclusion mechanism, Donnan exclusion/equilibrium and dielectric exclusion (electrostatic interactions) [13,14]. They include most organic molecules, viruses, and salts. In particular, NF membranes can reject divalent ions and are often used for water softening [15]. In contrast to MF, UF, and NF membranes, RO membranes possess extremely small pores (0.3–0.6 nm), therefore, they can be considered to be non-porous. Molecular transport in RO membranes is governed by a solution diffusion mechanism in which solutes are dissolved into the membrane material and diffuse along with the concentration gradient [16]. The separation is performed in different solubility and diffusibility of the solutes. The most common application of RO membranes is the desalination of brackish groundwater and seawater [17,18].

Ceramics and polymers are the two main materials for various membranes. In particular, ceramic membranes have gained more attention recently, due to their superior performance, hydrophilicity, mechanical robustness, and high thermal and chemical stabilities, which allow at least double lifespan compared with the polymer membranes [20,21]. The common ceramic materials used in membrane applications are Al_2_O_3_, TiO_2_, ZrO_2_, SiO_2_ [19,22], and those containing a combination of them such as Al_2_O_3_-ZrO_2_ [23] and TiO_2_-SiO_2_ [24], and various metal nanoparticles embedded in ceramics such as Ag-TiO_2_ [25]. However, the relatively high production cost of the ceramic membranes is a restricting parameter in widening of their applications. Indeed, polymeric materials are still more widely used, although there is a steadily decrease in the overall market share. Polymer membranes have the merits of being low cost, tunable in porous structure, and ease in scale-up. Therefore, the polymer membranes have been dominantly employed for water, wastewater treatment, and desalination [1]. Among them, Poly-ethersulfone (PES) [26], Poly-sulfone (PSf) [27], poly-vinylidene fluoride (PVDF) [28,29], poly-vinylpyrrolidone (PVP) [30], poly-acrylonitrile (PAN) [31,32], poly-vinyl alcohol (PVA) [33,34], and poly-vinyl acetate (PVAc) [35] are widely used as the polymeric membranes (Table 1). In addition to the poor life span, most of these polymeric membranes are inherently hydrophobic to certain extent, leading to low water flux, high fouling tendency, which often causes even shorter lifetime and higher operating cost.

One of the recent trends in membrane development is the ceramic-polymer composite membranes, which are able to integrate the advantages of ceramics and polymer materials, providing improvement in membrane performance for efficient separation and additional functionalities, while maintaining low cost. For example, the low flux and high fouling tendency due to hydrophobicity of polymer membranes could be mitigated by the addition of certain hydrophilic ceramic materials. Moreover, the material cost of ceramic-polymer composite membranes would be lower than that of ceramic membranes, since only a small amount of ceramic materials is included in the composite membranes. The concept of ceramic-polymer nanomembranes was reported in the 1970s. Ceramic fillers (SiO_2_, Al_2_O_3_, etc.) were mixed into cellulose acetate (CA) membranes [36]. Since the 1980s, ceramic-polymer composite membranes have been applied for gas separation [37,38,39], where the ceramic fillers enhanced the membrane rigidity for better separation.

In this overview, we first examine the different types of inorganic-organic composite membranes and then recent progress of these ceramic-polymer composite membranes for water treatment, in three categories: (i) Ceramic fillers in polymer (nanocomposite) membranes, (ii) thin film nanocomposite (TFN) membranes, and (iii) Ceramic-supported polymer membranes (Figure 2). The technologies to improve the compatibility of the two components, which is the key to successfully fabrication of high performance and stable ceramic-polymer composite membranes, are discussed. Finally, the perspectives and challenges for the future development of the ceramic-polymer composite membranes will be visited.

## 2. Composite Membranes

Composite membranes are commonly defined as polymer membranes, into which inorganic nanoparticles are incorporated, on which inorganic nanoparticles are deposited or which are supported by ceramic substrates as shown in Figure 2. There has been a steady rise in the published papers on the ceramic-polymer composite membranes, which have been employed in a large number of applications. More recently, new types of composite membranes are emerging, including those with MOFs (Metal-Organic Frameworks)/COFs (Covalent-Organic Frameworks) being incorporated/integrated. Moreover, there are also the composite-type membranes containing three/multi components, such as ceramic-(different) ceramic-polymer, GO-CNT-polymer, carbon-ceramic-polymer MOF-carbon-polymer etc.

### 2.1. Advantages of Inorganic-Polymer Composite Membranes

Simultaneous enhancement of flux and rejection

In general, flux and rejection are of the trade-off relationship. When large pores are created in a membrane in order to improve the flux, this leads to reduce the rejection. Both enhancement of flux and rejection can be achieved by the composite membranes. For example, Nambikkattu et al. improved both glucose flux and rejection simultaneously by addition of MgFe_2_O_4_ on PSf membrane [40], where the addition of MgFe_2_O_4_ enhanced hydrophilicity of the membrane, but did not influence the pore structure, thus resulting in an improvement of both flux and rejection simultaneously.

2.Addition of antibacterial and photocatalytic properties

Certain metal nanoparticles, such as Ag and Cu, are well-known for their antibacterial properties. Thus, by incorporation of Ag and Cu nanoparticles (NPs) into polymeric matrix, antibacterial properties can be added to the polymeric membranes. Indeed, high antibacterial behavior against *E. coli*, Bacillus Subtilis, etc. was reported in Ag and Cu NPs-containing polymeric membranes [41,42,43]. The superior antibacterial properties were also observed in GO-added PA membrane [44]. Likewise, photocatalytic property can be added into polymeric membranes by incorporation of TiO_2_ and ZnO [45,46,47]. Polymeric membrane themselves would not exhibit these antibacterial and photocatalytic properties.

3.Modification of morphology of polymeric membrane

The hydrophilicity of polymeric membrane can be enhanced by addition of certain hydrophilic inorganic NPs. Other properties can be also influenced by the addition of inorganic components into polymeric membranes. In particular, the morphology of polymeric components such as roughness, porosity, pore size, etc. can be strongly influenced, resulting in improvement of membrane performance [48,49,50].

4.Improvement of mechanical properties and thermal stability

The addition of certain inorganic component into polymeric membrane can improve the mechanical properties and thermal stability. For example, Poly-vinylidene chloride membrane with MCM-41 demonstrated higher tensile strength and then those of pristine polymeric membrane [51]. This phenomenon is more obvious in polymeric membrane on ceramic support [52]. Additionally, rGO addition to PDMS membrane increased the decomposition temperature of the polymer composite membrane [53].

### 2.2. Disadvantages of Composite Membranes

Compatibility between inorganic and organic components

One of the major issues with the nanocomposite membranes is the low compatibility between polymeric and inorganic components, which can cause aggregation of inorganic particles and separation of the two components, leading to poor performance, particularly in membrane stability. The stability of inorganic components in polymeric matrix has been performed by leaching test, exclusively in metal NPs-polymer membranes [54,55,56]. By contrast, in ceramic-polymer composite, the stability has been usually examined by the long-term antifouling test [57,58,59]. In the past decade, great progress has been made with improving the compatibility between both components, and a range of composite membranes with high stability have been prepared. Challenges in improvement of the compatibility between ceramic and polymeric components are summarized in Section 4.

2.Manufacturing cost

For polymeric membranes, phase inversion and casting have been widely used. These techniques can be directly applied for the nanocomposite-type membrane using polymer solution including inorganic NPs. In the case of TFN-type membrane, a polymeric support layer can be prepared first and the top inorganic layer is deposited on the support layer. In the case of ceramic-supported polymeric membrane, porous ceramic support is prepared first by sintering and then polymeric top layer is deposited by techniques such as dip coating, spin coating etc. These extra processes lead a degree of increase in membrane fabrication cost.

## 3. Ceramic-Polymer Composite Membrane

As shown in Figure 2, ceramic-polymer composite membranes can be divided into three groups. Nanocomposite membranes are composed of a polymeric membrane in which inorganic NPs are dispersed. This type of membranes has been most widely researched. The preparation for the nanocomposite membrane is mostly-based those well developed for polymer membranes, such as the phase inversion or casting of a polymer solution containing ceramic NPs. Either flat sheet or hollow fiber configurations can be obtained. The nanocomposite membrane has been used for both MF and UF processes. In the TFN membranes, a thin nanocomposite membrane is supported on a polymeric support, where ceramic NPs are located on the surface of the membrane and provide minimal influence on the intrinsic properties of polymeric substrate such as the pore structure. The surface properties of the resultant membrane are basically governed by ceramic NPs. The ceramic-supported polymer membranes consist of a thin polymer layer on a porous ceramic support. In contrast to the other two types of membranes, relatively dense and bulk ceramics, not ceramic NPs, are used in this type of membranes. The high chemical and thermal stability of ceramic supports restrict swelling of the thin polymer layer and improve flux and provide long membrane life. Highly tunable pore distribution and pore size of the polymeric surface layer influence the rejection properties of composite membranes. Coating on a polymer solution or in situ polymerization on a ceramic support has been employed to prepare the ceramic-support membranes. In all three types of membranes, not only the intrinsic properties of ceramic and polymeric components but also the interface properties between them influence the membrane performance significantly.

### 3.1. Ceramics in Polymer (Nanocomposite) Membranes

This type of membranes is composed of polymeric membranes in which ceramic NPs are dispersed in. Incorporation of the ceramic NPs into polymers could influence not only the hydrophilicity, pore size and distribution, surface roughness, but also can add new properties such as photocatalytic properties, antibacterial properties, etc. [60]. The fabrication of these membranes is mainly performed by casting and phase inversion (PI) using a polymer solution containing ceramic NPs [61].

Metal oxides such as SiO_2_, Al_2_O_3_, TiO_2_, Fe_3_O_4_ have been exclusively used as ceramic fillers for the nanocomposite membranes, where TiO_2_ is one of the most widely used ceramics in this type of membranes (Table 2). Additionally, natural minerals such as kaolin [62], cloisite [63] and montmorillonite [64] are studied to reduce material cost for inorganic components. The main advantages of TiO_2_ incorporation include the enhancement in hydrophilicity as well as antibacterial behavior by photocatalytic properties of TiO_2_ [65]. By the addition of TiO_2_, a decrease in contact angle and improvement of water flux have been reported by several groups. Additionally, UV-radiation enhances fouling resistance and antibacterial capability of TiO_2_-nanocomposite membranes due to the superhydrophilicity and photocatalysis of TiO_2_ under UV irradiation [66,67]. The UV irradiation also promote flux recovery of the TiO_2_-nanocomposite membrane [68].

The enhancement in hydrophilicity by the addition of ceramic NPs has also been observed in other transition metal oxides, such as SiO_2_ [81,82,83,84,85,86], Al_2_O_3_ [87,88,89,90], Fe_3_O_4_ [91,92,93,94,95], and ZrO_2_ [96]. The influence of these ceramic fillers on the properties of polymeric membranes is dependent on the type and amount of fillers being added. For example, the addition of mesoporous silica into PES UF membrane does not affect the pore size significantly, but increases the level of porosity, resulting in an improved water flux (Table 3) [85]. Contrarily, the addition of Fe_3_O_4_ into PES membrane largely influences both pore size and the level of porosity (Table 4) [94]. The level of porosity increased by the Fe_3_O_4_ addition, while the pore size drastically decreased.

Arsuaga et al. had compared the properties of TiO_2_, Al_2_O_3_, ZrO_2_-added PES UF membranes [96]. These ceramic fillers decreased the contact angle and enhance the level of porosity. The enhancement of water flux was in order of Al_2_O_3_ > TiO_2_ > ZrO_2_ (Table 5).

The effects of ceramic fillers in polymer membranes are significantly influenced by the properties of base polymeric membranes [83]. Table 6 summarizes the properties of various TiO_2_-nanocomposite membranes. For example, in the CA (cellulose acetate)-based nanocomposite, the contact angle is slightly increased by the addition of TiO_2_, while a decrement in contact angle is observed in PES, PPESK (poly-phthalazine ether sulfone ketone), and PSf-based composites. The TiO_2_ addition impacts the level of porosity of PES, PPESK, and PSf membranes, although it does not affect the level of porosity of PVDF membrane significantly. The ceramic fillers must be chosen carefully depending on the properties being improved.

### 3.2. Thin Film Nanocomposite (TFN) Membranes

This type of membranes is composed of a thin nanocomposite membrane supported on polymer substrates. The concept of TFN membrane was first suggested in the 1970s, and it has been widely studied for desalination of seawater/brackish water, removal of heavy metals, organic micropollutants and pharmaceutically active compounds [61]. PSf has widely been used as a supporting layer while PA (polyamide) has been widely employed in the thin top layer. As inorganic compounds, (a) metal oxide NPs, (b) metal NPs and (c) carbon materials such as CNT and GO have been studied (Table 7).

As in the nanocomposite membranes, TiO_2_ nanoparticles are among the most used in the thin top layer, and the water flux and antifouling properties are improved [49,100,101,102]. An optimum TiO_2_ loading in this type of membrane is reported as 0.05~0.1 wt.%. This is much lower than that in TiO_2_-nanocomposite membranes, which is around 1 wt.%. In the TFN configuration, TiO_2_ content can be drastically reduced. Exposing TiO_2_ NPs on the surface significantly influences surface properties of membranes, resulting in reduction of TiO_2_ content. The TFN membrane could be fabricated at lower material cost compared to the nanocomposite membrane. In contrast, photocatalytic properties of TiO_2_ in the TFN membranes have not been widely reported. The optimum ceramic NP content is also in the same range (0.05–0.1 wt.%) in the case of SiO_2_ NPs (Table 8) [103].

Like the nanocomposite membranes, metal NPs [115,116] and carbon materials [117,118,120,121,122,124,125,126,127,128,129,130,131,132,133,134,135,136,137,138,139] have been studies as additives for the TFN membranes. In particular, the unique properties of carbon materials (CNT, GO, rGO, etc.) and their derivatives influence membrane properties drastically even with small amounts of addition. 

### 3.3. Ceramic-Supported Polymer Membranes

This type of membrane is composed of a thin polymeric film (selective layer, active layer) supported on a ceramic porous substrate (Figure 3, [140]). The ceramic substrates provide the superior chemical, mechanical and thermal stabilities as well as negligible transport resistance and defines the external shape of the membrane [44]. The ceramic-supported polymer composites have attracted much attention for their significant performance in UF [141], pervaporation [44,133,134,135,136,137,138,139,140,141,142,143,144,145,146,147], gas separation [148], etc. The thin polymeric layer, which can consist of one or more intermediate layers, is prepared by processes, such as interfacial polymerization, dip coating, etc. The level of air humidity during dip coating, drying process and polymer solution affected quality of top thin layer significantly [148,149].

Table 9 summarizes the ceramic-supported polymer nanocomposite membranes reported. For the ceramic substrates, Al_2_O_3_, SiO_2_, TiO_2,_ and ZrO_2_ have been extensively used in forms of tubular, monolith, hollow fiber, and flat sheet. Additionally, natural minerals such as clay and kaolin have been applied to reduce the cost of ceramic substrates. Several types of polymeric thin films such as hydrophilic PA, PVA, PVP, PVAc, Chitosan, and hydrophobic polydimethylsiloxane (PDMS) have been employed for the thin top layer.

The benefits offered by the ceramic-supported polymer membranes are mainly in high flux and long-term stability. For example, PDMS/Al_2_O_3_-ZrO_2_ nanocomposite membrane shows about two times higher pervaporation flux of ethanol/water than that of the PDMS/Blend cellose acetate (BCA) membrane [163] (Figure 4). Additionally, the separation factor decreased with temperature monotonically in the PDMS/BCA membrane, while the peak of separation factor was shown at 50 °C in the PMDS/Al_2_O_3_.

Additionally, their superior long-term stability are reported by several research groups [148,150,159,163], where it is considered to be arising from the high structural, thermal, and chemical stabilities of ceramic supports in operating condition. In the polymer-supported membranes, swelling of the polymeric support in the operation could damage the thin top layer, resulting in poor stability of the polymer-supported membranes. The swelling tendency is more prominent at high temperatures. Therefore, most of the commercial PA membranes with polymer support cannot be used above 50 °C. Contrarily, PA supported on Al_2_O_3_ tubular membrane demonstrated stable rejection and permeation of MgCl_2_ at 70 °C, due to the high stability of Al_2_O_3_ support which could stabilize the PA top layer [152].

It must be noted that the swelling process is different between the polymer-supported and ceramic-supported polymer membranes (Figure 5) [163]. In the polymer support, top and support layers are swollen in a parallel direction together (Figure 5a). The swelling influences pore structure and membrane performance. On the contrary, only the top layer can be swollen in the ceramic-support membranes (Figure 5b). The ceramic support maintains its pore structure and can suppress the swelling of top polymeric layer. Therefore, influence of the swelling is reduced in the polymer-supported membranes. This would be one of the reasons behind the superior performance of the ceramic-supported composite polymer membranes.

Ceramic-supported polymer composite membranes can recover their performance completely by back washing after fouling [158]. For example, PDMS/β-Sialon membrane is fouled by the crystallization of NaCl. After the membrane is scoured and dried to remove the crystallized NaCl on the surface, the flux could be completely recovered. Interestingly, for example, Menne et al. reported a reusable Al_2_O_3_ monolith support for PDADMAC/PSS (poly(sodium 4-styrene sulfonate)) film [157]. After fouling, the top PDADMAC/PSS layer is removed by sodium hypochlorite (NaOCl) treatment. Then, a new top layer is built by the coating on the same Al_2_O_3_ monolith. Pure water permeability does not change by the removal and rebuilding of the top layer. This reusable ceramic support is expected to reduce material and production costs drastically.

By properly matching the properties between the ceramic support and a polymeric top layer, the ceramic-supported polymer composite membranes can feature high permeability. Additionally, the confinement in swelling of the polymeric top layer by a stable ceramic support provides long-term stability and allows high temperature operation. However, further research would be required to optimize the polymer-ceramic interface, in order to tailor the high performance of the ceramic-supported polymer composite membranes.

## 4. Strategies to Fabricate Ceramic-Polymer Composite Membranes

One of the challenges in fabrication of ceramic-polymer composite membranes is the incompatibility of the ceramic-polymer interface. According to the basic thermodynamic principles, the poor interfacial compatibility between ceramic and polymeric components would lead to separation and severe aggregation of each component, leaching out of ceramic nanoparticles, reduction in mechanical strength of the membranes, degradation in pore structure, change in hydrophilicity, and decrease in the stability of the composite membranes. In particular, some of the incorporated ceramic particles are usually highly polarized because of abundant polar moieties on their surfaces, while some commercial polymer membranes, such as PE and PP have a nonpolar nature. To address these issues, modification of both components is widely conducted to improve the compatibility of the ceramic-polymer interfaces. By an appropriate modification, the undesired differences between the ceramic particles and polymer matrix can be reduced. Surface modification can also improve membrane performance, such as increase in flux and fouling resistance [167,168]. The type of modifications is dependent on the membrane structure. In the nanocomposite membranes, ceramic surface and/or whole polymeric components can be tailored, while in the TFN membranes, mainly the surface modification of polymeric support is performed. Herein, some strategies to improve the compatibility between ceramic and polymeric components are described.

### 4.1. Modification of Ceramic Nanoparticles

In this strategy, ceramic NPs are modified with certain functional moieties for better compatibility with polymeric matrix. Intrinsically, cationic transition metals, such as Ti^4+^ in TiO_2_, can bond to the oxygen-containing functional groups such as COOH and SO_2_OH in the polymer matrix through coordination, stabilizing ceramic NPs in the polymer matrix [169]. However, in order to further improve the stability of the ceramic-polymer interface, the modification of ceramic NPs has been studied. For modifications of ceramic NPs, there are mainly three strategies, (1) surface modification, (2) surface functionalization, and (3) organic grafting on the particle surfaces.

#### 4.1.1. Surface Modification

Hydroxyl groups on the surface of ceramic NPs can form hydrogen bonds with carboxyl groups, amino groups, hydroxyl groups, etc. in the polymeric matrix, resulting in stabilization of the NPs (Figure 6) [170]. Wang et al. applied a hydrothermal treatment to grow gibbsite (Al(OH)_3_) on the surface of γ-Al_2_O_3_ NPs and then added them into the PVDF membrane [171]. The gibbsite can form hydrogen bonds with fluoride atoms in the PVDF membrane. The gibbsite/γ-Al_2_O_3_ NPs were stable for at least two days in dead-end filtration of MilliQ water at the flux of 5 × 10^−6^ to 2.5 × 10^−5^ m s^−1^. In addition, bio-fouling by *E. coli* could be greatly reduced. The stabilization of ceramic NPs in the polymeric matrix by the interaction between surface hydroxyl group on the ceramic NPs and functional groups in the polymeric matrix was also observed in MnO_2_ [172], ZnO, and ZrO_2_ [173] as well.

Surface sulfonation of inorganic particles has been also performed because the surface sulfonate group can react with the amide group in the polymetric matrix, leading to a high stability of the nanocomposite membrane [174]. For example, Sun et al. incorporated 3-mercapto-propyltrimethoxysilane-modified H-ZSM-5 zeolite into a chitosan membrane [175]. The SO_3_H group could be easily grafted on the surface of H-ZSM-5 and interacted with NH_3_^+^ group in the chitosan membrane. By the interaction, not only leaching of ceramic NPs, but also the creation of nonselective voids at the interface between H-ZSM-5 and chitosan membrane could be suppressed, resulting in a high separation factor of 274.46 in the pervaporative dehydration of aqueous ethanol solution.

Additionally, the surface SO_3_H group on ceramic NPs can increase the negative charge of the nanocomposite membrane and enhance protein rejection due to electrostatic repulsion with negatively charged proteins. For example, sulfonated-TiO_2_ (S-TiO_2_) composited with PES UF membrane showed a higher bovine serum albumin (BSA) rejection as well as the level of porosity and hydrophilicity than those of pristine PES and non-sulfonated TiO_2_ composite membranes (Table 10) [176].

The surface modification of ceramic NPs can provide not only an improvement in compatibility with polymeric matrixes but also influences on surface charge and pore structure of polymeric matrixes, resulting in better performance of the composite membranes. However, the relationship between the modification and influence on the pore structure is still unclear in several cases.

#### 4.1.2. Surface Functionalization on Ceramic Nanoparticles

Surface functionalization by in situ generation/growth of an inorganic component is also an effective approach to reduce the agglomeration and provide additional functionality for ceramic particles in polymer matrix [177]. For example, Zhang et al. coated SiO_2_ on Fe_3_O_4_ particle by the hydrolysis of tetraethyl orthosilicate [178], where the surface hydrated SiO_2_ layer prevented the aggregation of Fe_3_O_4_ NPs in PES polymer matrix and interacted with oxygen atom in the PES matrix. The membrane could be operated for 168 h, continuously, in which the effect of coating of hydrated SiO_2_ is similar to surface hydration of ceramic NPs. However, hydrophilic SiO_2_ possesses more hydroxyl groups on the surface, giving rise to a higher stability in the polymeric matrix compared with direct surface modification of Fe_3_O_4_ NPs. Same strategy is applied for ZrO_2_-coated SnO_2_ [179], MCM-41-coated SrCo_x_Cu_1-x_O_3-λ_ [180], and TiO_2_-coated hollysite nanotube [181].

The surface functionalization can change the stability of ceramic NPs in the polymeric matrix drastically compared with the surface modification. The functionalization is not limited to oxides and hydroxides. Recently, superior performance of CuS-coated CuO in PVDF was reported [182]. The superior performance could be observed in 60 min. Although long-term stabilization must be examined, the surface functionalization by other compounds such as sulfide and phosphide is a new research area.

#### 4.1.3. Organic Grafting

Polymeric matrixes are highly compatible with polymer components. The introduction of the grafted polymer chain on ceramic NPs shall increase the stability of the NPs in the polymeric matrix and suppresses aggregation of ceramic particles (Figure 7). For example, a hydrophilic polymer, poly (2-hydro-xyethyl methacrylate) (p(HEMA), was grafted on TiO_2_ nanoparticles. The surface p(HEMA) group helps improve the dispersibility of TiO_2_ nanoparticles in PSf flat UF membrane [183]. Compared to unmodified TiO_2_, the p(HEMA)-grafted TiO_2_ enhances hydrophilicity and pure water flux and reduces BSA UF resistances. Similar research on polymer grafting was performed using various polymer including zwitter ionic polymers (Table 11). In all these studies, aggregation of inorganic NPs is suppressed and hydrophilicity and water flux of the composite membranes are enhanced.

The introduction of certain grafted polymer chains onto ceramic NPs clearly shows improving dispersity and compatibility of ceramic NPs with polymer matrix, resulting in enhancement in performance of the nanocomposite membranes. By grafting various types of polymers, not only tailoring the properties but also adding on one or more new functions to the nanocomposite membranes. However, the introduction of a complicated functional group on the surface of ceramic NPs needs rather complicated processing steps, making membrane production difficult and costly. Finding effective and simple functional groups and the development of novel functionalizing processes would be desirably required.

### 4.2. Modification of Polymers

Another direction to improve the compatibility between ceramic and polymeric components is the modification of polymeric matrixes by introducing functional groups that can interact with ceramic NPs. Polar groups such as sulfonic and carboxyl groups can provide active sites to capture on the surface of ceramic components as well as an increase in hydrophilicity [191,192,193].

#### 4.2.1. Modification of Bulk Polymer Matrix

PVDF contains fluorine atoms, which exhibit a high electronegativity, in its polymer chain. Therefore, a negatively charged fluorine atom and positively charged adjacent hydrogen atom can contribute the hydrogen bond to surface hydroxyl groups on certain ceramic NPs. To further stabilize ceramic NPs, grafting of other polymers containing polar groups has been performed. For example, Zhang et al. grafted poly(acrylic acid) into PVDF membrane [97]. Carboxyl groups in poly (acrylic acid) (PAA)-grafted PVDF can interact with Ti(OBu)_4_ that is a precursor for TiO_2_. As a result, TiO_2_ NPs are uniformly dispersed in the PAA-grafted PVDF composite membrane. A similar study is performed using CA as a polymer and TEOS (tetraethyl orthosilicate) as a precursor for SiO_2_ [194] and poly-dopamine (PDA)-poly-ethyleneimine (PEI)-g-poly-acrylonitrile (PAN) as a polymer and Zr(SO_4_)_2_ as a precursor for ZrO_2_ [195]. PDA is also employed to immobilize Ag NPs in PSf UF membranes [196,197].

Adding a component into the polymeric matrix is also an effective method, but adding an appropriate polymer does not require the polymerization like the grafting mentioned above. Thus, the functional polymeric matrix can be embedded easily. For example, mixing the triblock copolymer PEO-PPO (poly-phenyleneoxide)-PEO, which contains moieties that can interact with both organic and inorganic components, into PES UF membrane can improve the interfacial compatibility with TiO_2_ NPs [198]. As other additives, PDA (polydopamine) has been exclusively researched [198,199,200]. In this strategy, many polymers with moieties which can interact with both polymeric and ceramic components, can be good candidates as the additives. There are still a lot of room to develop suitable additives in this strategy.

#### 4.2.2. Surface Modification of Polymer Matrixes

In the TFN membrane, ceramic NPs are supported on polymer matrixes. Therefore, surface modification of polymer matrixes has been extensively studied in order to prepare stable TFN membranes. The modifications have been largely performed to solid polymers, rather than the polymer solution. For example, γ-methacryloxy propyl trimethoxy silane (MPTS) monomer is polymerized (grafted) onto the PVDF solid membrane and then hydroxyl-rich TiO_2_, SiO_2_ and β-FeOOH are introduced [201]. The ceramic NPs were then firmly anchored in the grafted poly-MPTS layer and the ceramic NPs were stable even under ultrasonication for 30 min. The same effect was observed in (3-aminopropyl)-triethoxysilane grafted PVDF with SiO_2_ NPs membrane [202] and trimesoyl chloride which reacts with Si-OH groups, grafted PVDF membrane [203]. Additionally, functional complexes such as polyoxometalate could be immobilized onto the surface modified polymer membranes [204].

Recently, in order to create oxygen-containing functional groups on the polymeric membrane surface, plasma treatment is applied. For example, PP (Polypropylene) membrane was treated with the O_2_ plasma followed by the dip coating of TiO_2_ NPs [205]. By the O_2_ plasma treatment, the C=O stretching band was formed on the PP membrane, which facilitates stabilizing TiO_2_ NPs. Additionally, the oxygen-containing functional groups were prepared on PVDF membrane by Ar plasma treatment, followed by oxygen exposure [206]. The functional groups facilitated to graft PAA thin layer on the PVDF membrane. The PAA layer helped immobilize TiO_2_ NPs by the coordination of Ti^4+^ to carboxylic groups. As a result, the membrane possesses high hydrophilicity, water flux, and oil rejection rate. In another example, negatively charged PMAA (Polymethacrylic acid) chains are created onto the PVDF membrane surface via plasma-induced grafting polymerization [207]. Carboxyl moieties on PMAA acted as the binding sites to attract amino-grafted SiO_2_ NPs. In general, the plasma treatment is able to give rise to stable composite membranes. Nonetheless, further research is needed to clarify the exact effect of the plasma treatment in the performance of nanocomposite membranes.

### 4.3. Direct Deposition of Ceramic NPs in Polymer Matrixes

In situ growth of inorganic components on membrane surfaces cannot only improve the compatibility between the two components, but also provide ease of tailoring the inorganic layers. This strategy was first proposed by Xu et al. [208,209,210], where the creation of an intermediate layer on polymer membrane surface could capture precursors of inorganic particles, which are then converted into inorganic NPs on the membranes. The observed difference from the modification of the polymer surface described in the previous section is attributed that surface modified-polymer captures precursors of ceramic NPs, not directly captures ceramic NPs.

For example, PAA brushes are grafted on PP membrane, where the PAA-PP membrane is soaked into CaCl_2_ solution and then Na_2_CO_3_ solution. The COO^−^ groups in the PP brushes captures Ca^2+^ followed by CaCO_3_ formation through reaction with Na_2_CO_3_, resulting in CaCO_3_ deposition on the membrane (Figure 8) [211]. The PAA intermediate layer could not only provide binding sites for CaCO_3_ growth but also stabilize the amorphous CaCO_3_ to control the mineral layer thickness. In the follow-up works, it was found that the PDA/PEI layer could provide positive amino groups for silicification and catechol groups for chelating metal ions. Rigid hydrophilic ceramic coatings showed excellent anti-oil properties in water, enabling these to be used in oil-in-water emulsion separation [212,213,214]. Such properties are also desirable in Li-ion battery separators to improve electrolyte wetting and resist thermal shrinkage [215].

Since the PDA-based interlayers being used, controlled growths of nickel-cobalt layered double hydroxides on a PDA-modified PVDF [216] and ZrO_2_ on PDA/PEI [195] have been studied. Compared with a single intermediate PDA layer, an additional positively charged CS layer could overcome the partial congregating of PDA and served as a smooth platform for uniform in situ growth of SiO_2_, leading to a dense and defect-free SiO_2_ layer [217]. Other groups have also studied this strategy [218,219,220,221].

The in situ growth of inorganic NPs supported by grafting functional polymer components can form rather uniform and stable inorganic particles, leading to superior membrane performance. The popularization of this technique largely relates to the simplification in grafting processes of functional polymer components and the in situ growth process of inorganic components. For the in situ growth process, a moderate heating is sometimes performed, and however, the process could cause a slight change in the level of porosity and pore sizes of the polymer component. The grafting and growing processes must be chosen carefully.

As an alternative method for in situ growth of inorganic components on polymer membranes, surface coating techniques can be applied. Among the surface coating techniques, the Atomic Layer Deposition (ALD) technique is the most widely studied for the deposition of inorganic particles on polymeric membranes [222]. In a typical ALD process, reactive precursor vapors of inorganic materials are pulsed into a chamber alternately under the protection of inert gas, leading to the layer-by-layer growth of metals, metal oxides, and even organic materials [223,224]. The deposition of a series of oxides ZnO, Al_2_O_3_, TiO_2_, and SnO_2_ on PVDF membranes, can be performed by the ALD and tested anti-crude-oil properties (Figure 9) [225]. The ALD would be a good choice to prepare TFN membranes because the ALD can deposit various inorganic layers with a controllable thickness. However, for certain extremely inert substrate materials (e.g., PP or PTFE), ALD coating becomes difficult and an effective surface treatment is needed [226,227,228]. The TFN membranes prepared by the ALD technique is summarized in a review [222]. The application of the ALD technique for membrane preparation is a relatively new topic. Further intensive research is needed.

### 4.4. Ceramic-Supported Polymer Membranes

For the ceramics-supported polymer membranes, in addition to interfacial adhesion, penetration of polymer solution into pores of ceramic support can inhibit the formation of a dense and defect-free polymeric separation layer on the ceramic support. To prevent the penetration, an increase in viscosity of polymer solution and pre-wetting pores of ceramic support has been performed [223]. The pores of ceramic support are filled with water first and then polymer solution was cast on the ceramic support. Because of the blocking effect of the filled water, the penetration of polymer solution can be greatly reduced. The interfacial adhesion strength between the polymeric layer and ceramic supports must be high enough to ensure long-term operation of the ceramic-supported polymeric membrane. The interfacial adhesion is largely influenced by the roughness of ceramic support and viscosity of polymer solution [229]. An increase in the adhesion strength was obtained by lowering viscosity and roughing the support surface. For example, Jin’s group developed a home-made in situ nano-indentation/scratch technique and studied critical load on the interfacial adhesion of the PDMS/Al_2_O_3_ membrane [230]. Recently, tubular Al_2_O_3_ substrate was dip-coated into PEI solution short time (~10 s) to prevent the penetration of the polymer solution and then phase-inverted from both inner and outer surface of the substrate [231]. In this case, the penetration could be suppressed completely.

By contrast, interfacial polymerization can also be employed to prepare a thin polymeric layer directly on the ceramic support. For example, Chong et al. prepared a PA thin layer with a thickness of 30–40 nm on the Al_2_O_3_ tubular membrane by interfacial polymerization [153]. Xia et al. also succeeded in fabricating PA thin layer on the Al_2_O_3_ tubular UF membrane by interfacial polymerization [151]. The interfacial polymerization is suitable for tubular support which is difficult to be cast polymer solutions. Shi et al. deposited a PEI gutter layer on tubular Al_2_O_3_ support by dip coating, which enhances the adhesion of the PDMS top layer [232]. The third component must have high compatibility with both the ceramic support and the thin polymeric top layer.

As an alternative strategy, the silane grafting approach reveals enhancement of the interfacial adhesion and performance of ceramic-support polymer membranes for ultra-filtration of oil-water emulsions [233] and pervaporation [150]. In this approach, methoxysilane with polymer chains reacts with hydroxyl groups on the ceramic surface. The Si-O covalent improve surface adhesion with ceramic support and the polymer chains enhance compatibility with the polymeric top layer (Figure 10).

## 5. Perspective

Ceramic-polymer composite membranes have been intensively researched in recent years. To successfully develop the designed types of composite membranes, different fabrication techniques have been established, especially to improve compatibility between ceramic and polymeric components. Tuning the overall membrane structure can dramatically change the physicochemical properties, such as the type and level of porosity, surface hydrophilicity, chemical and mechanical stabilities, which impact the overall filtration/separation performance. Additional functionalities such as antibacterial, photocatalytic, and antifouling properties can also be anchored in ceramic-polymer composite membranes. Well-designed ceramic-polymer composite membranes would be able to solve current problems of polymeric and ceramic membranes and be a good candidate for next generation membrane technology.

To further develop into practical applications of the composite membranes, there are several perspectives and challenges, which are summarized as follows:Fundamental study: Fundamental understandings for the influence of certain ceramic NPs on membrane structures and membrane performance is still unclear, in several cases. There have been several studies reporting influence of ceramic NPs on polymeric matrixes; however, the origin of the influence has not been researched in detail. As shown in Table 4, the influence of TiO_2_, Al_2_O_3_ and ZrO_2_ on the level of porosity and contact angle of polymeric matrix were different. However, it is still unclear how and what properties of the ceramic NPs influence the polymeric matrix. Detail studies on a relationship between surface properties of ceramic NPs, structures and performance of the membrane would be needed. This will help design new composite membranes.Structural study: Structural studies of the composite membranes have mostly been carried out by observation of morphology using SEM and sometimes by EDS, FT-IR and XPS. These rather simple structural analyses can lead to a wrong conclusion. For example, Fe-Boemite-PVB/PVDF membrane was prepared by (i) casting Boehmite-PVDF-PVB solution, (ii) immersing Boehmite-PVB-PVDF membrane into FeSO_4_ solution, and (iii) Reduction of Fe^2+^ on the Boehmite-PVB-PVDF membrane by KBH_4_ [234]. In the structural analysis of the membrane made by using SEM and FT-IR, the TFN-type membrane structure was envisioned. In the solution casting, the boemite particles could exist not only surface of the PVDF-PVB membrane, inside the PVDF-PVB membrane as well. Fe^2+^ ion would be captured by OH groups of boehmite inside the polymeric matrix. Thus, the membrane would be concluded as nanocomposite-type membrane, not TFN-type. The lack of structural analysis of membrane could cause misunderstanding of membrane structure, leading to the wrong conclusions.Long-term stability: there are several research works conducted in improving the compatibility between ceramic NPs and polymeric matrix, as described. However, knowledge of long-term compatibility is still not enough. In short term studies, there have been suggestions to suppress the aggregation of NPs in composite membranes and nanoparticle leakage. Studies on long-term compatibility between ceramic NPs and polymeric matrix, change of the membrane properties and environmental impact by the nanoparticle leakage would be required. Particularly, in addition to photocatalytic properties by TiO_2_, photocurrent would oxidize and damage/change the polymeric matrix. This should become more prominent after any long-term usage. However, studies on extended long-term stability of the photocatalytic membranes have not been performed. To understand the stability of the membranes and fate of nanoparticles, several characterizations, e.g., leaching tests, scanning electron microphotographs of the membrane surface and cross-section, roughness and FTIR with attenuated total reflectance (ATR) scans of used membranes must be carried out.Production cost: the expected application of the ceramic-polymer composite membranes for water treatment is still at a rather early stage. There are numerous laboratory-based works, but studies on large-scale production and industrial application have not been properly conducted [235]. More efforts have to be made to evaluate the long-term durability under the application conditions and cost-effectiveness including the supply of nanoparticles and methods for nanoparticle incorporation. Compared to current polymetric membranes, composite membranes require additional production processes, leading to higher production costs. More recently, novel fabrication techniques such as 3DP are emerging for membrane fabrication. Some of the new techniques have been applied for ceramic and polymeric membranes so far [236,237,238]; however, there were fewer for ceramic-polymer composite membranes. The 3DP would be able to reduce the production cost for the composite membranes, when properly developed. Give the Al_2_O_3_ membrane is several times more expensive than the PES polymeric membrane [164], development of cost-effective new ceramic NPs and the corresponding preparation process of composite membranes would be needed. Natural polymers, such as cellulose acetate and polysaccharide, normally possess polar groups in their structures, which can provide interacting sites for ceramic particles [239,240]. Additionally, the material cost of natural minerals such as kaolin, natural clay, etc. is much lower than Al_2_O_3_ [167]. Additionally, these natural minerals possess high hydrophilicity, which is expected to improve the hydrophilicity of composite membranes, if they are incorporated into the polymeric matrix. The usage of natural polymers and natural minerals would be a strategy to reduce the production cost of the composite membranes. Further study must be performed to reduce the production cost of the nanocomposite membranes.

## Figures and Tables

**Figure 1 molecules-26-03331-f001:**
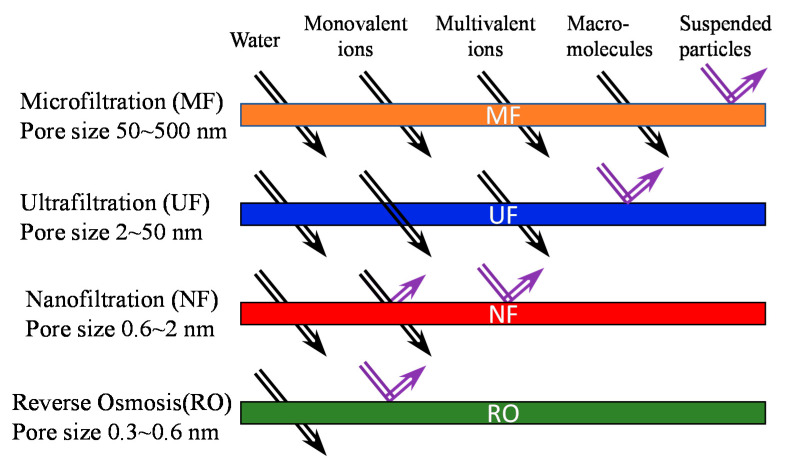
Pore size ranges of various membranes and the ability to reject particular foulants [19]. Reproduced with permission. Copyright in 2019, Elsevier.

**Figure 2 molecules-26-03331-f002:**
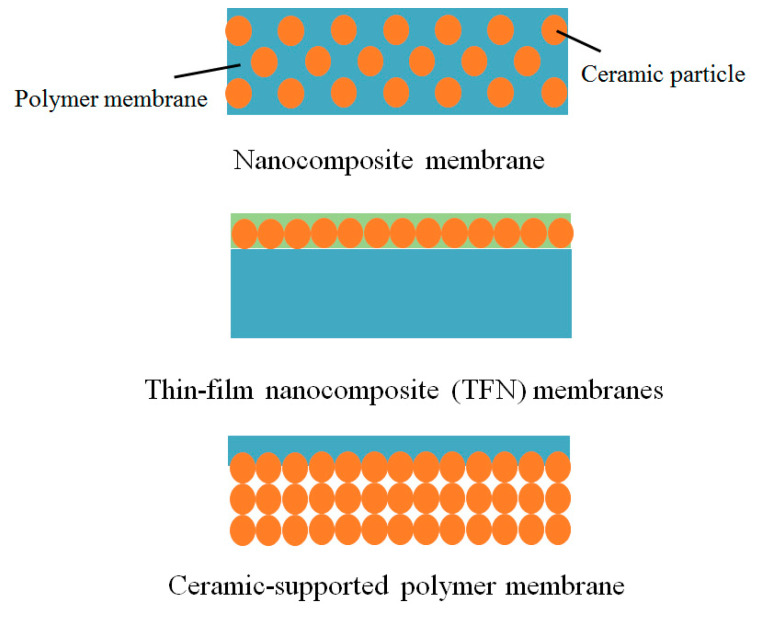
Structures of ceramic-polymer composite membranes.

**Figure 3 molecules-26-03331-f003:**
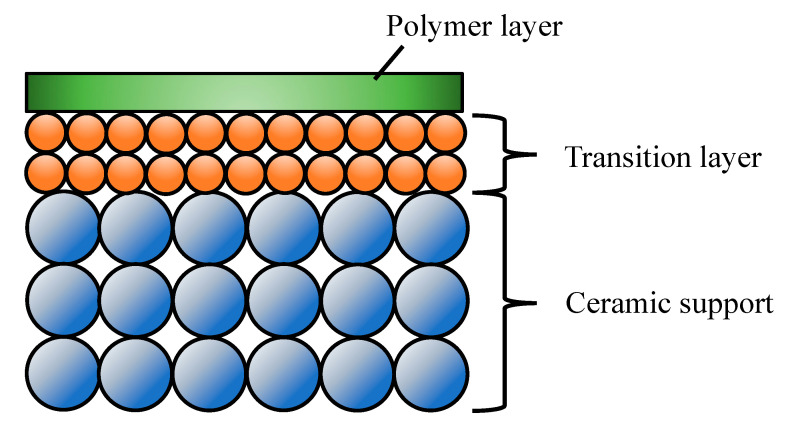
Structure of the ceramic-supported polymer composites.

**Figure 4 molecules-26-03331-f004:**
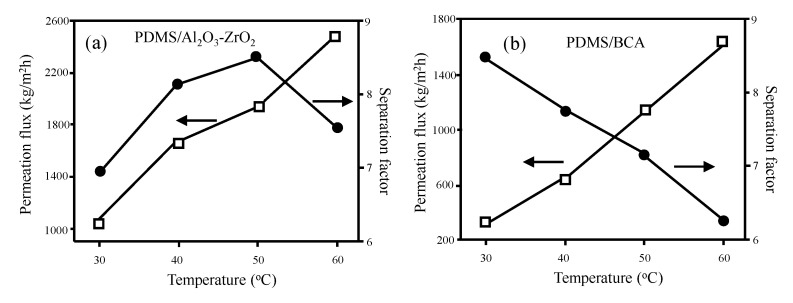
A comparison of pervaporation performance between (**a**) PDMS/Al_2_O_3_-ZrO_2_ and (**b**) PDMS/BCA membranes [163]. Reproduced with permission. Copyright in 2011, Elsevier.

**Figure 5 molecules-26-03331-f005:**
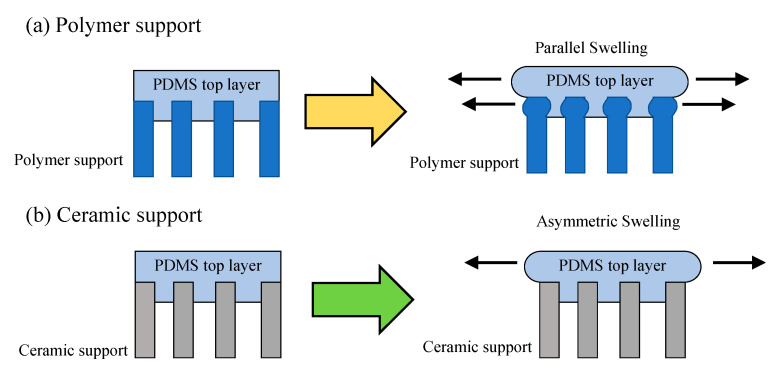
Swelling in (**a**) polymer-supported PDMS top layer and (**b**) ceramic-supported PDMS top layer [163]. Reproduced with permission. Copyright in 2011, Elsevier.

**Figure 6 molecules-26-03331-f006:**
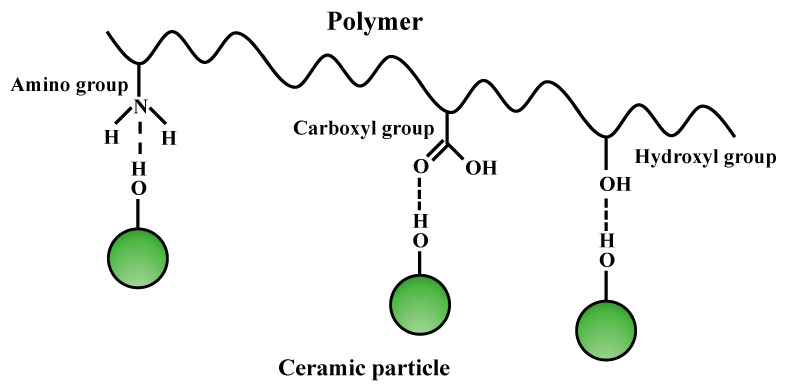
Illustration of the interfacial hydrogen bonds between hydroxyl group on the ceramic particles and functional groups in the polymeric matrix.

**Figure 7 molecules-26-03331-f007:**
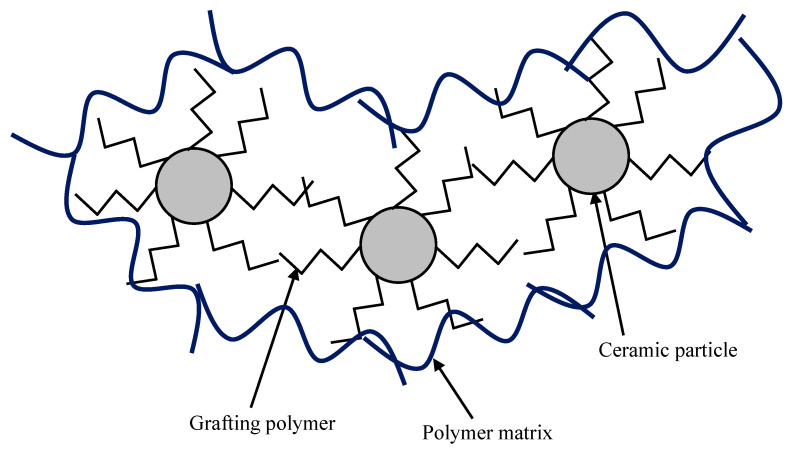
Schematic illustration of composite membrane using graft polymer chain on ceramic NPs. The grafting polymer suppress aggregation of the ceramic NPs and interacts with polymer matrixes, improving stability of the ceramic NPs in the polymer matrixes.

**Figure 8 molecules-26-03331-f008:**
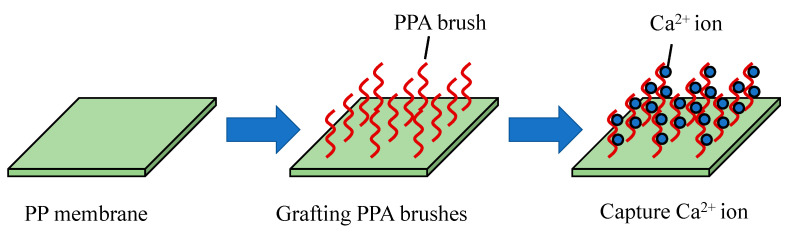
Preparation of CaCO_3_ mineral-coated membrane.

**Figure 9 molecules-26-03331-f009:**
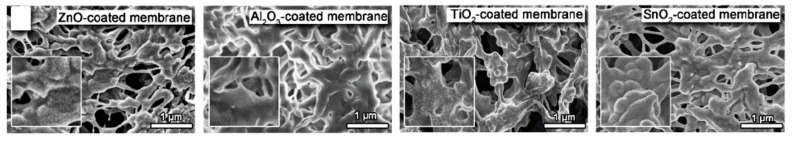
ZnO, Al_2_O_3_, TiO_2_ and SnO_2_ deposited on PVDF membrane by ALD [225]. Reproduced with permission. Copyright in 2018, American Chemical Society.

**Figure 10 molecules-26-03331-f010:**
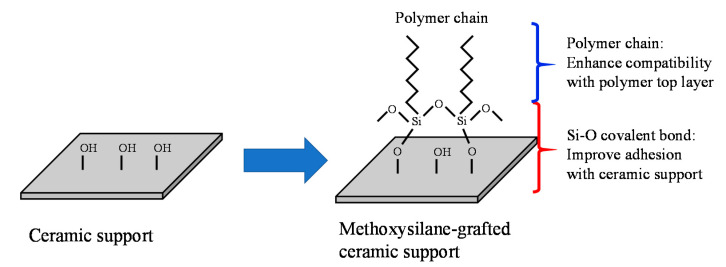
Fabrication of PDMS membrane supported on surface-grafted ceramic substrate.

**Table 1 molecules-26-03331-t001:** Structures of common polymer membranes.

Polymer	Abbreviation	Structure
poly-acrylonitrile	PAN	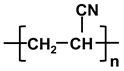
poly-vinylpyrrolidone	PVP	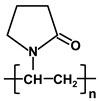
poly-vinylidene fluoride	PVDF	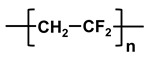
poly-vinyl alcohol	PVA	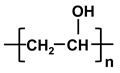
poly-vinyl acetate	PVAc	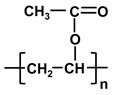
Poly-ethersulfone	PES	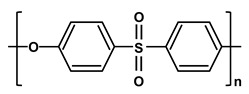
Poly-sulfone	PSf	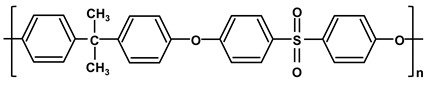

**Table 2 molecules-26-03331-t002:** Summary of TiO_2_-based nanocomposite membranes.

Ceramics	Polymer	Membrane	Improved Properties by Addition of Filler	Ref.
TiO_2_ (25 nm)	PVDF	Hollow fiber UF	Hydrophilicity	[68]
TiO_2_	PVDF	Hollow fiber UF	Hydrophilicity	[28]
TiO_2_ (25 nm) + dopamine	PVDF + PVP	UF	Hydrophilicity, Photocatalycity	[69]
TiO_2_ (20 nm)	PES	UF	Hydrophilicity, antifouling performance	[70]
TiO_2_ (20 nm)	PVDF	MF	antifouling performance	[71]
TiO_2_	PES	NF	Permeability, antifouling performance	[72]
TiO_2_ (21 nm)	PSf	Hollow fiber UF	antifouling performance	[73]
TiO_2_ (62 nm)	CA	-	Thermal stability, Water flux	[74]
TiO_2_ (20 nm)	PVDF-sulfonated-PES	Flat sheet UF	Hydrophilicity, Antifouling, photo-bactericidal effect	[75]
TiO_2_ (20 nm)	PVDF	UF	Hydrophilicity, antifouling performance	[76]
TiO_2_ (20 nm)	PVDF	UF	Hydrophilicity, Photocatalycity	[77]
TiO_2_	PAN	-	Mechanical properties, Hydrophilicity, Photocatalycity	[78]
TiO_2_	PVDF	-	Oil rejection, Water flux	[79]
TiO_2_	PSf	-	Dye removal	[80]

**Table 3 molecules-26-03331-t003:** Porosity, pore size and water flux of SiO_2_-contained PES membrane [85].

Membrane	Porosity (%)	Pore Size (nm)	Water Flux (L m^−2^ h^−1^ at 200 kPa)
PES	69.2	12.9	122
1wt% SiO_2_-PES	74.8	13.7	145
2wt% SiO_2_-PES	75.9	14.6	180
4wt% SiO_2_-PES	74.9	12.8	137

**Table 4 molecules-26-03331-t004:** Level of porosity, pore size and water flux of Fe_3_O_4_-contained PES membrane [94].

Membrane	Porosity (%)	Pore Size (nm)	Water Flux (L m^−2^ h^−1^ at 200 kPa)
PES	17	89.3	5
1wt% Fe_3_O_4_-PES	33	40.1	6
5wt% Fe_3_O_4_-PES	44	3.9	12
10wt% Fe_3_O_4_-PES	59	5.8	21

**Table 5 molecules-26-03331-t005:** Contact angle, level of porosity and water flux of TiO_2_, Al_2_O_3_, ZrO_2_-added PES membrane [96].

Membrane	Contact Angle (°)	Porosity (%)	Water Flux (L m^−2^ h^−1^ at 300 kPa)
PES	52.3	51.8	182
TiO_2_-PES	44.1	66.6	199
Al_2_O_3_-PES	37.8	62.1	209
ZrO_2_-PES	48.6	64.3	190

**Table 6 molecules-26-03331-t006:** Properties of various TiO_2_-added membranes reported.

Membrane	Contact Angle (°)	Porosity (%)	Ref.
PES	52.3	51.8	[96]
0.4 wt.% TiO_2_-PES	44.1	66.6	[96]
PPESK	50.7	80.5	[97]
1 wt.% TiO_2_-PPESK	45.9	86.8	[97]
PSf	70.1	63.4	[98]
1 wt.% TiO_2_-PSf	52.0	81.0	[98]
CA	69.3	-	[99]
5 wt.% TiO_2_-CA	71.1	-	[99]
PVDF	-	72.2	[54]
1 wt.% TiO_2_-PVDF	-	73.9	[54]

**Table 7 molecules-26-03331-t007:** Summary of surface nanocomposite membranes.

Top Layer	Polymeric Support	Improved Properties	References
TiO_2_-PI	PES	Salt rejection, Water flux	[100]
Halloysite nanotube-PA	PSf	Antifouling performance, Water flux	[101]
TiO_2_-PA	PSf	Antifouling performance	[102]
Fe_3_O_4_/ZnO-PA	PSf	Hydrophilicity, Water flux	[48]
SiO_2_-PA	PSf	Hydrophilicity, Permeability, Salt rejection	[103]
Al_2_O_3_-PA	PSf	Antifouling performance, Water flux	[104]
TiO_2_/Halloysite nanotube-PA	PSf	Antifouling performance, Recovery	[49]
NaY-PA	PSf	Hydrophilicity, Permeability	[50]
Mesoporous-silica-PA	PSf	Antifouling performance, Water flux	[105]
ZnO-PDMS	PI	UV resistance, Superoleophilicity	[106]
Clay-Chitosan	PVDF	Dye adsorption	[107]
TiO_2_	PS	Hydrophilicity, Water flix	[108]
TiO_2_	PAN	Water flux, Dye rejection	[109]
SiO_2_	PVDF	Dye rejection, Oil rejection	[110]
Bi_12_O_17_Cl_2_	CA	Dye removal	[111]
MoS_2_	PVDF	Salt rejection, Dye rejection	[112]
ZnWO_4_	PVDF	Dye rejection	[113]
TiO_2_	CA	Water flux, Dye rejection	[114]
Au-Ag-PAA	PA	Antifouling and antibiofouling performance	[115]
Arginine-Fe-PA	PES	Antifouling performance, Permeability	[116]
CNT-PA	PSf	Antifouling performance, Salt rejection	[117]
Amine-MWCNT-PA	PSf	Permeability, Salt rejection	[118]
MWCNTs-PA	PSf	Antifouling performance, Water flux	[119]
GO-PA	PAN	Antifouling performance, Hydrophilicity	[120]
PVP-GO-PA	PSf	Salt rejection, Water flux	[121]
GO/Fe_3_O_4_-PA	PES	Water flux, Antifouling performance	[122]
GO-PA	PSf	Water flux, Slat rejection	[123]
GO-PA	PSf	Water flux, Hydrophilicity	[124]
rGO-PDMS	PES	Thermal stability	[125]
Quantum dot graphene-PA	PES	Anti-bacterial property, long-term stability	[126]
Fullerenol (C_60_(OH)_n_)-PA	PSf	Hydrophilicity, antifouling property	[127]

**Table 8 molecules-26-03331-t008:** Separation properties of PA/PSf membranes with various SiO_2_ contents [103].

Membrane	Pure Water Permeability (L m^−2^ h^−1^ bar^−1^)	NaCl Rejection (%)
PA/PSf	2.94	72
0.01 wt.% SiO_2_-PA/PSf	5.88	82
0.05 wt.% SiO_2_-PA/PSf	9.52	89
0.1 wt.% SiO_2_-PA/PSf	12.36	78

**Table 9 molecules-26-03331-t009:** Summary of the ceramic-supported polymer membranes.

Polymer Thin Film	Ceramic Support	Improved Properties	References
PDMS	Al_2_O_3_ hollow fiber	Butanol/water separation factor, long-term stability	[150]
PA	Al_2_O_3_ tubular UF membrane	Dye rejection, methanol permeability	[151]
PA	Al_2_O_3_ tubular	Salt rejection, water permeability	[152]
PDMS	ZrO_2_/Al_2_O_3_	Sulfur removal efficiency	[153]
PVA	Fumed silica	Water selectivity, Pervaporation separation index	[52]
Melamine-terephthaldehyde	Al_2_O_3_	n-heptane permeability, dye rejection	[154]
PDADMAC/poly(sodium 4-styrene sulfonate	Al_2_O_3_ monolith	Stability for backwashing, reusable ceramic support	[155]
PA	Al_2_O_3_ hollow fiber	Water flux	[156]
Poly (maleic anhydride-alt-1-alkenes)	γ-Al_2_O_3_ NF membrane	Dye rejection, permeability	[157]
Sulfonated polybenzimidazole	TiO_2_, TiO_2_/ZrO_2_ tubular	Mechanical ruggedness, flux	[148]
PDMS	β-sialon	Long-term stability, recovery ability	[158]
PDMS	ZrO_2_/Al_2_O_3_ tubular	High flux, recovery ability	[159]
PA	Al_2_O_3_	High H_2_/CO_2_ selectivity	[160]
PVP	ZrO_2_	Oil rejection, anti-fouling performance	[161]
Chitosan	Clay+Kaolin	Rejection of mercury and arsenic, cost of membrane	[162]
PDMS (polydimethylsiloxane)	ZrO_2_/Al_2_O_3_ tubular	High flux, Membrane stability	[163]
PVAc, PVP	Al_2_O_3_ tubular	Separation factor	[164]
PVAc	SiO_2_ tubular	Water flux	[165]
PDMS	γ-Al_2_O_3_	IPA selectivity	[166]
PVA	ZrO_2_/Al_2_O_3_ tubular	Water permeability, Selectivity of water to ethyl acetate	[167]
PSf-PEI	Flat Pozzolan	Water permeability, Dye rejection	[168]
Sulfonated polybenzimidazole	TiO_2_/ZrO_2_ tubular	Flux, Pervaporation stability	[149]

**Table 10 molecules-26-03331-t010:** Selected properties of PES, non-sulfonated TiO_2_/PES and S-TiO_2_/PES membranes.

Membrane	Porosity (%)	Contact Angle (°)	BSA Rejection (%)
PES	68.4	75	88
Non-sulfonated TiO_2_/PES	77.3	60	92
S-TiO_2_/PES	87.6	49.2	99

**Table 11 molecules-26-03331-t011:** Grafted ceramic NPs-polymer membranes.

Grafted Ceramic NPs	Host Polymer	Reference
PHEMAb-PMMA@-SiO_2_ ^(1)^	PVDF	[184]
silylated ZSM-5	PDMS	[57]
Polyester-MWCNT	PVDF	[185]
PEG-GO	PVDF-co-HFP	[186]
PMMA-CNT	PVDF	[58]
PEG-nanodiamond	CA	[187]
PMMA-SiO_2_	PES	[188]
P(PEGMA)-GO	PSf	[189]
PSBMA-MoS_2_ ^(2)^	PES	[190]

^(1)^ PHEMAb-PMMA: poly(hydroxyethyl methacrylate)-block-poly(methyl methacrylate). ^(2)^ PSBMA: 2-methacryloyloxy ethyl dimethyl (3-sulfopropyl)-ammonium hydroxide sulfobetaine methacrylate.

## Data Availability

The data presented in this review paper are available in article.

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
