# Peer review of "Ceramic-Polymer Composite Membranes for Water and Wastewater Treatment: Bridging the Big Gap between Ceramics and Polymers"

_molecules, 2021, doi:10.3390/molecules26113331_

Round 1
Reviewer 1 Report
This review paper thoroughly overviews three types of ceramic-polymer composite membranes. And by looking into strategies to improve the compatibility among ceramic and polymeric components, the manuscript conclude the perspectives and challenges for the future development of the composite membranes. This review maybe provide great interest and inspiration to broad readers about ceramic-polymer composite membranes. But before this manuscript can be considered for publication, the following questions need to be addressed.
1. In the Abstract,“in the past decade, great progress has been made in improving the compatibility between ceramics and polymers, while the synergy between them has been among the main pursuits, especially in the development of the high performing nanocomposite membranes for water and wastewater treatment at lowered manufacturing cost”suggested to be deleted.
2. The classification of ceramic-polymer composite membrane((i) ceramics in polymer membranes, (ii) thin-film nano-composite (TFN) membranes, and (iii) ceramic-supported polymer membranes) is not very clear, it is suggested to modify.
3. Where is Figure 3?
4. Fig. 2. 4, 5, 6, 7, 8,10 is too rough, please redraw it.
5. In table 8, suggests adding some data support in "Improved Properties”.
6. Explain the mechanism of Figure 5 in detail
7. “3.1.2. Surface functionalization” and “3.2.2 Surface functionalization” are same.
8. In Figure 9, if there are any Reference cited to? If so, please add it.
9. It is suggested to add the latest references in recent years
10. Finally, please modify the format of the manuscript
Author Response
- In the Abstract,“in the past decade, great progress has been made in improving the compatibility between ceramics and polymers, while the synergy between them has been among the main pursuits, especially in the development of the high performing nanocomposite membranes for water and wastewater treatment at lowered manufacturing cost”suggested to be deleted.
Thank the reviewer for the valuable suggestion. The sentence was deleted.
- The classification of ceramic-polymer composite membrane((i) ceramics in polymer membranes, (ii) thin-film nano-composite (TFN) membranes, and (iii) ceramic-supported polymer membranes) is not very clear, it is suggested to modify.
Thank the reviewer for the valuable suggestion. To clarify the classification of membrane structure types, Figure 2 is added.
- Where is Figure 3?
Figure 3 (Figure 2 in the revised manuscript) is added.
- Fig. 2. 4, 5, 6, 7, 8,10 is too rough, please redraw it.
Those figures have been redrawn.
- In table 8, suggests adding some data support in "Improved Properties”.
Thank the reviewer for the suggestion. Data (Fig. 4) and explanation is added.
- Explain the mechanism of Figure 5 in detail
More explanations for Figure 5 are added.
- “3.1.2. Surface functionalization” and “3.2.2 Surface functionalization” are same.
They have been changed into “4.1.2. Surface functionalization on ceramic nanoparticles”, “4.2.2 Surface modification of polymer matrixes”
- In Figure 9, if there are any Reference cited to? If so, please add it.
Yes, citation is added.
- It is suggested to add the latest references in recent years
Recent papers have been added.
- Finally, please modify the format of the manuscript
The manuscript has been formatted using the template.

Reviewer 2 Report
This is a review paper without scientific achievements. It should be presented in that way. Moreover, I would like to see the improving effect of the combined membranes.
I saw some formatting problems, e.g. t Figure 2 (maybe my reader made the problem.)
The article is too long.
Round 2
Reviewer 1 Report
Authors made considerable modifications in the manuscript as suggested by reviewers. Now the paper can be accepted for publication in the present form.
Reviewer 2 Report
I accept the revised version